# MULTITRUST: ENHANCING SAFETY AND TRUSTWORTHINESS OF LLMS FROM MULTIPLE SAFETY PERSPECTIVES

## ABSTRACT

Large Language Models (LLMs) have shown impressive performance across various tasks, yet they still face significant safety and trustworthiness challenges, such as robustness, fairness, and truthfulness. Addressing these challenges is critical for the reliable deployment of LLMs. Directly fine-tuning LLMs to enhance safety can degrade their performance and is challenging to balance across multiple safety perspectives due to the forgetting phenomenon. In this paper, we propose MultiTrust, a novel and scalable framework designed to enhance LLM safety from multiple safety perspectives. In particular, MultiTrust first generates challenging training data through adversarial optimizations, focusing on LLMs trustworthiness perspectives, such as robustness, fairness, and safety. MultiTrust then separately train safety auxiliary models for each perspective using supervised fine-tuning and Direct Preference Optimization (DPO). MultiTrust augments a base model with these safety auxiliary models on the fly through dynamic routing and logit ensembling, significantly boosting the performance across different trustworthiness metrics for the base model while preserving its helpfulness. Notably, MultiTrust introduces an effective perplexity-based inference-time router to seamlessly integrate these safety auxiliary models by averaging the logit outputs of the selected safety auxiliary model and the base model, which enhances the stability of the final performance. Moreover, MultiTrust's flexible design allows for the augmentation with new safety auxiliary models for different perspectives without necessitating additional training or adaptation. Extensive experimental results show that MultiTrust, which trains a series of 7B safety auxiliary models, significantly improves the trustworthiness of the base LLM across different sizes (7B and 13B). For instance, MultiTrust increased the average performance of Llama2-13B from 35.54% to 51.14%, and Vicuna-13B from 29.91% to 52.82%, outperforming models with similar and even larger sizes across different perspectives. These results underscore the effectiveness and scalability of MultiTrust in enhancing the safety and reliability of LLMs.

## 1 INTRODUCTION

Large Language Models (LLMs) have demonstrated unprecedented capabilities in a wide range of applications, setting new benchmarks for complex tasks and frequently achieving human-like proficiency in various domains (OpenAI, 2022; Wang et al., 2023b). Despite these advancements, significant concerns about their safety and trustworthiness persist (Wang et al., 2021; 2023a; Mazeika et al., 2024). For instance, LLMs often exhibit vulnerabilities in different trustworthiness perspectives such as robustness, fairness, privacy, and truthfulness, which can severely limit their deployment in sensitive or safety-critical environments (Driess et al., 2023).

Traditionally, approaches to enhance model safety have addressed these issues in isolation, focusing on optimizing one perspective of trustworthiness at a time, typically through continuous fine-tuning on domain-specific datasets. While this approach may resolve isolated issues, it often meets the "model forgetting" phenomenon, where improvements in one trustworthiness perspective may inadvertently cause performance degradation in others (Dou et al., 2023). Furthermore, such sequential

training can reduce the overall benign performance or helpfulness of the models, limiting their deployment in real-world scenarios (Dou et al., 2023).

In contrast to methods that sequentially enhance various perspectives of model trustworthiness, some existing work has attempted to integrate multiple functionalities into a single framework (Ilharco et al., 2022; Bansal et al., 2024). However, these approaches generally lack flexibility; they are tailored to specific base models and configurations, which is hard to adapt to diverse base models and emerging new perspectives. Such frameworks typically require extensive fine-tuning or the training of additional modules, making them computationally expensive and less scalable (Bansal et al., 2024).

In response to these challenges, we introduce MultiTrust, a novel framework that treats different trustworthiness perspectives as distinct functionalities, allowing for dynamic model safety alignment based on the specific needs of each task. MultiTrust proposes a unique alignment technique that dynamically aligns the output of a base LLM with the output of specialized safety auxiliary models. This is achieved through a perplexity-based inference-time router, which intelligently selects the most appropriate safety model during inference based on the input query, and combines the logits from the base and safety models, thereby improving the trustworthiness of the base LLM without necessitating additional training or fine-tuning.

We have conducted extensive experiments and demonstrated the effectiveness of MultiTrust in significantly enhancing the trustworthiness of LLMs. By applying our framework to align base models of different sizes, we enhance their performance across different trustworthiness perspectives, without compromising their general helpfulness. For instance, MultiTrust elevated the average performance score of Llama2-13B from $35.54\%$ to $51.14\%$ and Vicuna-13B from $29.91\%$ to $52.82\%$, surpassing both similar and larger-sized models. We also observe interesting findings, for example, we find that learning from the differences between answers using DPO is more effective than SFT. We also notice that datasets that focus on different trustworthiness perspectives may interact with other perspectives, helping to improve the model performance in other perspectives.

Our contributions are threefold: 1) **Challenging Data Generation.** We provide novel methods to generate data that challenge model robustness using various adversarial attack algorithms, and create fairness-sensitive datasets with balanced representation across sensitive attributes. 2) **Flexible Model Safety Alignment.** Our perplexity-based router enables the seamless integration of new trustworthy models without additional training or fine-tuning, aligning model outputs by combining logits from the base and selected safety auxiliary models. 3) **Enhanced Trustworthieness Performance.** MultiTrust consistently improves the performance of a given base LLM across different trustworthiness perspectives, demonstrating its effectiveness, flexibility, and scalability in enhancing the safety and trustworthiness of LLMs.

## 2 METHOD

Addressing the safety and trustworthiness of Large Language Models (LLMs) presents several critical challenges. For example, retraining or extensively fine-tuning the base model to enhance safety can be prohibitively expensive and time-consuming. To circumvent this, MultiTrust trains additional, specialized safety auxiliary models rather than modifying the existing base LLM directly. This approach significantly reduces the computational overhead associated with continuously updating the base LLM. Besides, the need to flexibly consider different safety perspectives and adapt to new ones as they emerge presents a significant challenge. To address this, we propose a novel perplexity-based router that enables the seamless integration of new safety auxiliary models without additional training. Moreover, balancing the trade-off between maintaining good performance in benign scenarios and ensuring safety and truthfulness across various trustworthiness perspectives is challenging. We mitigate such trade-offs by proposing an effective logit-based alignment approach that aligns the base LLM with different safety auxiliary models flexibly. The pipeline of our MultiTrust is shown in Figure 1, which consists of 3 components, 1) creating challenging data for different trustworthiness perspectives, 2) training a safety auxiliary model for each trustworthiness perspective, and 3) base LLM alignment with safety auxiliary models. In this section, we will introduce the details of different components, demonstrating how to obtain safe and trustworthy LLMs given a pretrained base LLM.

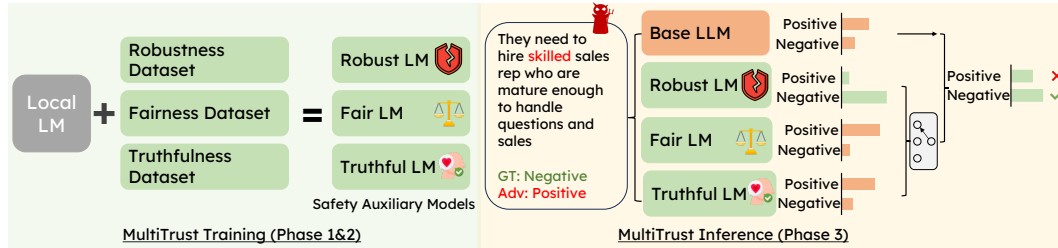

Figure 1: **Overview of MultiTrust.** MultiTrust consists of 3 different phases. In the first phase, we create challenging data for different trustworthiness perspectives. In the second phase, we train different safety auxiliary models for different perspectives based on the dataset collected in the first phase. In the last phase, we align the base LLM with selected safety auxiliary models by aggregating the logits of the models. The resulting aligned LLM is safe and trustworthy in different perspectives.

## 2.1 GENERATING CHALLENGING DATA FOR DIFFERENT TRUSTWORTHINESS PERSPECTIVES

MultiTrust is a general framework that can be leveraged to enhance the model performance in different perspectives. Here we mainly focus on the following 3 trustworthiness perspectives: adversarial robustness, fairness, and safety, following the categorization provided in existing benchmarks (Liang et al., 2022; Wang et al., 2023a).

**Adversarial robustness.** To defend the adversarial attacks and make our model more robust to input manipulations, we identify the vulnerabilities in existing models and collect our challenging dataset by adding adversarial perturbations to existing datasets. Specifically, we generate diverse adversarial data against LLMs (Wang et al., 2021), and consider the following five most representative and challenging tasks: Sentiment Analysis (SST-2), Duplicate Question Detection (QQP), and Natural Language Inference (NLI, including MNLI, RTE, QNLI). We sample the training sets of these tasks, reformulate the data into the instruction-following format, and perform word-level adversarial attacks to generate our challenging robustness dataset. In our experiments, we consider the following five kinds of word-level perturbations: typo-based perturbation (TextBugger (Li et al., 2018)), embedding-similarity-based perturbation (TextFooler (Jin et al., 2020)), context-aware perturbation (BERT-ATTACK (Li et al., 2020)), knowledge-guided perturbation (SememePSO (Zang et al., 2019)), and semantic-optimization-based perturbation (SemAttack Wang et al. (2022)) to perform white-box attacks against Llama2 (Touvron et al., 2023) model. These attacking strategies are originally designed to attack BERT-like models, which have different classification protocols from GPT-like models. We propose to adapt and modify the adversarial optimization process and use the conditional probabilities of (adversarial) candidate labels given the prompt to optimize the adversarial sentences. After optimizing the input prompt, we collect a ground truth answer and a wrong answer. For tasks with two labels, we select the two labels as the ground truth and the wrong answer, respectively. For tasks with three labels, such as MNLI, we select the opposite label as the wrong answer when the ground truth is 'yes' or 'no', and we randomly select 'yes' or 'no' as the wrong answer when the ground truth is 'maybe'. We will release our generated adversarial dataset for public evaluation.

**Fairness.** To develop a fair model and close the performance gap of the model on different groups with different sensitive attributes, we construct a fair distributed dataset by considering various sensitive attributes. Specifically, we leverage two commonly used fairness evaluation tabular datasets Adult (Asuncion & Newman, 2007) and Crime (Redmond, 2009), and transform the tabular data in these dataset into language descriptions. Each instance in the Adult dataset includes 14 attributes of a person (e.g., age and education level) as input. The task is to predict whether the income of the person is over $50k per year. We consider multiple sensitive attributes, including sex (white and black), race (male and female), and age (above average and below average). In the Crime dataset, each instance has 10 attributes of a community, such as education level and unemployment rate. We construct different queries based on controlled protected variables (e.g., demographic attributes). The protected variable in the Crime dataset is selected as the race (e.g., the portion of white in the community). We follow common practice (Wang et al., 2023a) to split the dataset, and we only sample data in the training split to avoid potential contamination. Based on these two datasets, we

construct a fair distributed training dataset. For each instance in the dataset, we first clone a duplicate and then flip the sensitive attribute of the cloned instance. In this way, we hope to reduce the model's dependence on these sensitive attributes and make more fair predictions. We convert the ground truth label to the true answer and use the opposite label as the wrong answer.

**Truthfulness and safety.** To encourage the model to generate truthful content, we leverage the GRATH dataset (Chen et al., 2024) to improve the truthfulness of our models. GRATH dataset is collected by iteratively prompting a pre-trained LLM to give a correct and an incorrect answer to a given question. The two answers are used to fine-tune the pre-trained LM. Then the fine-tuned model can be prompted again to generate pair-wise correct and incorrect answers of higher quality. In our experiments, we adopt the data collected by GRATH in the first two iterations.

## 2.2 TRAINING SAFETY AUXILIARY MODELS FOR DIFFERENT SAFETY PERSPECTIVES

Based on our generated challenging data for different safety perspectives, we independently train a distinct safety auxiliary model for each perspective. Our methodology comprises a two-phase training process. Initially, we conduct supervised fine-tuning (SFT) of the pre-trained LM on the collected datasets. Subsequently, we apply Direct Preference Optimization (DPO) to refine the performance by learning from the differences between correct and incorrect responses.

Formally, we define our dataset as $D := \{(q^i, a_T^i, a_F^i)\}_{i=1}^n$, where $q^i$ denotes the input question or prompt, $a_T^i$ the correct answer, and $a_F^i$ the incorrect answer. Let $\pi_\theta$ represent the safety auxiliary model to be optimized, initially derived from a pre-trained model $\pi_{\text{pre}}$. During the SFT phase, the model is trained to maximize the likelihood of the correct answer with the loss function defined as:

$$\mathcal{L}_{\text{SFT}}(\theta) = -\sum_{i=1}^n \log \pi_\theta(a_T^i \mid q^i).$$

In the subsequent DPO phase, the model is further refined by learning from the difference between the correct and incorrect answers. The DPO loss function is expressed as:

$$\mathcal{L}_{\text{DPO}}(\theta) = -\frac{1}{n} \sum_{i=1}^n \left[ \log \sigma \left( \beta \log \frac{\pi_\theta(a_T^i|q^i)}{\pi_{\text{ref}}(a_T^i|q^i)} - \beta \log \frac{\pi_\theta(a_F^i|q^i)}{\pi_{\text{ref}}(a_F^i|q^i)} \right) \right]$$

where $\sigma$ is the logistic function and $\beta$ is a parameter controlling the deviation from the base reference policy $\pi_{\text{ref}}$, which is fixed and initialized as the supervised fine-tuned model $\pi_{\text{SFT}}$.

## 2.3 ALIGNING BASE LLMS WITH SAFETY AUXILIARY MODELS

To integrate the base LLM $\pi_{\text{base}}$ with the trained safety auxiliary models, we introduce a perplexity-based routing mechanism that selects an appropriate safety auxiliary model during inference, based on the input question. This selection is critical for dynamically adapting to different safety perspectives with minimal computational overhead. This selection is made by evaluating the perplexity of the input with each model and choosing the model that minimizes it:

$$\pi^* = \underset{\pi \in (\pi_{\text{adv}}, \pi_{\text{fair}}, \pi_{\text{truth}})}{\arg\min} \exp\left( -\frac{1}{T} \sum_{t=1}^T \log \pi(q_t^i|q_{<t}^i) \right) \tag{1}$$

where $\pi_{\text{adv}}, \pi_{\text{fair}}, \pi_{\text{truth}}$ are the 3 safety auxiliary models trained in the previous phase. $\log \pi(q_t^i|q_{<t}^i)$ denotes the log-likelihood of the $t$-th token conditioned on the preceding tokens. $T$ represents the total number of tokens in the input question. Through the auxiliary model selection process in Equation (1), MultiTrust determines the best model to handle the current input question, by selecting the model with the lowest perplexity.

Following the selection of the optimal safety auxiliary model, we proceed to align the base LLM with the chosen safety auxiliary model by combining the logits from both models. The alignment is done through the equation:

$$\log \pi_{\text{trust}}(a_t^i|q^i, a_{<t}^i) \propto \log \pi_{\text{base}}(a_t^i|q^i, a_{<t}^i) + \gamma \log \pi^*(a_t^i|q^i, a_{<t}^i) \tag{2}$$

where $\pi_{\text{trust}}$ is the resulting trustworthy model and $\gamma$ is a weighting factor that balances the influence of the safety auxiliary model. By leveraging the inference-time alignment process in Equation (2),

the model can flexibly integrate the knowledge from safety auxiliary models. By introducing the parameter $\gamma$, we are able to balance the trade-off between trustworthiness performance and benign general task performance.

## 3 EXPERIMENTAL RESULTS

### 3.1 EXPERIMENT SETTINGS

**Models.** We fine-tune all our safe models based on Llama-2-Chat-7B Touvron et al. (2023), a widely-used open-source 7B LLM. We use the trained safe models to augment three different models with different sizes: Vicuna (Chiang et al., 2023) (7B and 13B) and Llama-2-Chat (13B). We compare MultiTrust against a variety of open-access models featured on LLM Safety Leaderboard (Wang et al., 2023a) and Open LLM Leaderboard (Beeching et al., 2023), with parameters sizes ranging from 7B to 33B, including Gemma-it (Team et al., 2024), Zephyr (Tunstall et al., 2023), Qwen-Chat (Bai et al., 2023), Mistral-Instruct (Jiang et al., 2023), Llama-3-Instruct (AI@Meta, 2024), and Vicuna (Chiang et al., 2023).

**Datasets.** To rigorously evaluate the effectiveness of MultiTrust, we leverage various datasets, focusing on both trustworthiness and general performance. Our selection of trustworthiness datasets includes: 1) **DecodingTrust** (Wang et al., 2023a): This safety benchmark is designed to assess different perspectives of model trustworthiness. For the purposes of our experiments, we specifically focus on the adversarial robustness and fairness sections to assess the corresponding capabilities of MultiTrust. 2) **TruthfulQA** (Lin et al., 2021): Designed to measure the truthfulness of a language model's responses, this dataset helps us evaluate how accurately MultiTrust generates truthful answers in response to various questions. Our selection of general tasks includes: 1) **AI2 Reasoning Challenge (ARC)** (Clark et al., 2018): a dataset of grade-school science questions; 2) **HellaSwag** (Zellers et al., 2019): a dataset for commonsense inference; 3) **MMLU** (Hendrycks et al., 2020): a dataset to measure a text model's multitask accuracy. 4) **Winogrande** (Sakaguchi et al., 2021): an adversarial and difficult Winograd benchmark at scale, for commonsense reasoning.

**Implementation details.** During robustness data generation, we adopt similar hyper-parameters used in different attack algorithms to attack a pre-trained Llama-2 model, resulting in $78,715$ adversarial examples across 5 different tasks. For fairness data generation, we sample $3,000$ data from Adult dataset and $1,000$ data from Crime dataset, which lead to $8,000$ data in total after balancing the sensitive attributes. The truthfulness dataset we leveraged contains $2,184$ examples generated using Llama-2 model. During safety auxiliary model learning, we first SFT the model on the corresponding dataset for 1 epoch, then run DPO on the answer pairs for $1,000$ steps. All experiments are conducted using A100 GPUs.

### 3.2 EFFECTIVENESS OF MULTITRUST

**MultiTrust is effective in enhancing LLM safety and trustworthiness.** We present the trustworthiness performances of various open-access models and MultiTrust on three different models in Table 1. This analysis distinctly showcases significant improvements across all three trustworthiness perspectives when comparing the base models to their MultiTrust-aligned counterparts. For example, for the Vicuna-13B model, we observe a marked increase in the robustness score from $30.50$ to $66.90$, and the fairness score from $16.22$ to $43.40$. Similarly, the truthfulness score for the Llama-2-13B model significantly rises from $35.99$ to $44.51$. These enhancements highlight MultiTrust's ability to significantly boost the trustworthiness of models without the need for additional training. Moreover, when MultiTrust-aligned models are compared with other open-access models, it becomes evident that MultiTrust not only enhances models of comparable sizes but also outperforms larger models. For instance, the MultiTrust-aligned Vicuna-7B model achieves an average trustworthiness score of $52.60$, surpassing the $42.07$ score of the much larger Vicuna-33B. More detailed results of each trustworthiness perspective can be found in Appendix A.

**MultiTrust introduces minimal impact on the model's helpfulness** In addition to trustworthiness assessments, we evaluated the general performance of our models on established benchmarks. The results are summarized in Table 1. This analysis reveals that the performance of the base models and their MultiTrust-aligned counterparts are similar, underscoring the effectiveness of MultiTrust in maintaining general performance while enhancing trustworthiness. Particularly, the results from the

Table 1: **Overall performance of MultiTrust on trustworthiness benchmarks and general helpfulness benchmarks.** We compare the performance of various open-access models with MultiTrust applied to three base LLMs. For each model, we report the performance on both trustworthiness and helpfulness. For trustworthiness, we report the scores on adversarial robustness, fairness, and truthfulness. For helpfulness, we report the scores on the ARC Challenge, HellaSwag, MMLU, and Winogrande. For MultiTrust, we show the size of the base LLM. All scores are the higher the better. We **bold** the highest score of each task and underline the second best.

| Model | Size | General performance | | | | Trustworthiness perspectives | | | |
|---|---|---|---|---|---|---|---|---|---|
| | | ARC | HellaSwag | MMLU | Winogrande | Adv | Fair | Truth | Avg |
| Llama-2 | 7B | 53.50 | 78.58 | 47.39 | 72.61 | 46.47 | 25.34 | 37.78 | 36.53 |
| Gemma | 7B | 50.85 | 71.79 | 51.80 | 67.80 | 43.43 | 27.65 | 38.71 | 36.60 |
| Zephyr | 7B | **63.65** | **84.31** | 59.86 | 77.66 | 27.83 | 29.80 | 46.84 | 34.82 |
| Qwen v1.5 | 7B | 56.31 | 78.60 | 60.15 | 67.64 | 45.93 | 42.76 | 44.92 | 44.54 |
| Mistral v0.1 | 7B | 55.38 | 75.48 | 53.71 | 75.45 | 32.47 | 43.56 | 47.58 | 41.20 |
| Llama-3 | 8B | 62.12 | 78.77 | **65.67** | 75.45 | 42.61 | 37.50 | 43.99 | 41.37 |
| Vicuna | 33B | 63.05 | 83.13 | 59.46 | **77.90** | 61.92 | 17.66 | 46.62 | 42.07 |
| Vicuna | 7B | 53.92 | 77.43 | 49.97 | 72.38 | 49.75 | 36.36 | 41.65 | 42.59 |
| MultiTrust Vicuna-7B | 7B | 53.58 | 75.63 | 50.12 | 68.90 | **67.07** | 43.73 | 46.99 | 52.60 |
| Llama-2 | 13B | 60.32 | 82.14 | 53.60 | 74.27 | 37.30 | 33.32 | 35.99 | 35.54 |
| MultiTrust Llama-2-13B | 13B | 57.94 | 78.44 | 52.23 | 71.98 | 64.86 | **44.06** | 44.51 | 51.14 |
| Vicuna | 13B | 57.34 | 81.18 | 55.70 | 75.45 | 30.50 | 16.22 | 43.00 | 29.91 |
| MultiTrust Vicuna-13B | 13B | 56.23 | 77.29 | 53.81 | 72.06 | 66.90 | 43.40 | **48.16** | **52.82** |

ARC and the MMLU dataset suggest that these benchmarks are less influenced by trustworthiness alignment. For instance, the accuracy on the ARC for the Vicuna-7B model shows a minor decrease from 53.92% to 53.58%. Similarly, in the MMLU benchmark, the Llama2-13B model experiences a slight reduction in accuracy from 53.60% to 52.23%.

## 3.3 IN-DEPTH ANALYSIS OF MULTITRUST

In this section, we conduct a comprehensive exploration and analysis of MultiTrust. We begin by evaluating the common practice of using SFT on trustworthy datasets. Our findings confirm that while SFT is a straightforward method, it typically leads to sub-optimal performance. Besides, when considering new safety perspectives, SFT usually leads to the model forgetting phenomenon, where improvements in one area may cause regressions in another. We proceed by comparing the impacts of SFT and DPO within our two-step safety auxiliary model learning process. We find that DPO is more helpful to enhance the model performance compared to SFT. And integrating both methods significantly improves the overall trustworthiness of the models. We then explore the inference-time routing mechanism introduced in MultiTrust. This part of our analysis focuses on how effectively the router selects the most appropriate safety auxiliary model based on the input query, thereby optimizing the model trustworthiness in the responses. Finally, we investigate the interactions between different trustworthiness perspectives within the framework. This exploration helps us understand how improvements in one aspect of trustworthiness may influence others, providing insights into the complex dynamics of model safety and reliability.

**Continue fine-tuning leads to model forgetting.** One intuitive approach to incorporate new trustworthiness perspectives into a model is to continue fine-tuning it on datasets corresponding to each perspective. However, our findings indicate that this approach results in significant model forgetting. In our experiments, we initially train the model on the robustness dataset, followed by sequential fine-tuning on fairness and truthfulness datasets. After each training phase, we evaluate the robustness performance of the model. The results of these evaluations are depicted in Figure 2, which also includes the baseline performance of the pre-trained Llama-2-7B-Chat model for reference. We observed an initial improvement in robustness scores following fine-tuning on the robustness data. However, subsequent fine-tuning on the fairness dataset led to a decline in robustness scores, and this decline continued as the model was further fine-tuned on the truthfulness dataset, eventually nearing the baseline performance of the pre-trained model. These results clearly demonstrate that sequential fine-tuning on different datasets causes the model to forget previously learned tasks.

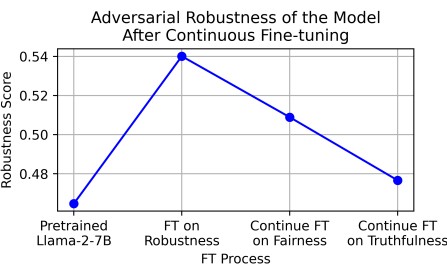

Figure 2: **Adversarial robustness of the base LLN after continuous fine-tuning.** We show the robustness of a pre-trained Llama-2-7B model after continuously fine-tuning on robustness dataset, fairness dataset, and truthfulness dataset, respectively. The forgetting phenomenon is clearly observed.

Table 2: **Trustworthiness comparison of models fine-tuned on a mixture of data and fine-tuned on different datasets separately.** We report the model performance on different trustworthiness perspectives. $FT_{mix}$ denotes the model fine-tuned on a large mixed dataset of three different trustworthiness perspectives. $FT_{seq}$ denotes the models fine-tuned on each dataset separately. Note that numbers in $FT_{sep}$ is obtained with 3 different models separately.

| Model | Adv | Fair | Truth | Avg |
|---|---|---|---|---|
| Llama-2-7b | 46.47 | 25.34 | 37.78 | 36.53 |
| $FT_{mix}$ | 45.97 | 31.89 | 36.55 | 38.14 |
| $FT_{sep}$ | 55.50 | 34.37 | 37.10 | **44.98** |

**Fine-tuning on dataset mixtures leads to sub-optimal performance across all datasets.** Another strategy to enhance model capabilities across multiple trustworthiness perspectives involves fine-tuning on a mixed dataset comprising various trustworthiness-focused datasets. However, our experiments shown in Table 2 indicate that this approach yields suboptimal results when compared to fine-tuning on individual datasets dedicated to specific perspectives. For instance, a model fine-tuned on a mixture of three different datasets achieved a score of 45.97 on the robustness perspective. In contrast, a model fine-tuned exclusively on the robustness dataset scored significantly higher, with a performance of 55.50 on the same perspective. Additionally, the average performance score of the model trained on the mixed dataset was 38.14, which falls below the average score of 44.98 achieved by models trained separately on each dataset. These findings underscore the limitations of generalized fine-tuning approaches and highlight the effectiveness of our safety auxiliary model training methodology.

**Learning from the difference between answers improves model trustworthiness** In the safety auxiliary model training phase of MultiTrust, we consider a two-step learning process to enhance model trustworthiness. Initially, we employed Supervised Fine-Tuning (SFT) to train the model using the correct answers from the dataset. Subsequently, we employed Direct Preference Optimization (DPO) to further refine the model's ability to discern between correct and incorrect responses. We compare our combined method with two methods separately. The results are shown in Table 3. Our experimental findings reveal that DPO training substantially outperforms SFT in enhancing the trustworthiness of the models. For example, the DPO-trained model gets 64.78 averaged score across three perspective, while the SFT-trained model only gets 55.50. This improvement is likely attributable to the DPO's focus on learning from the contrasts between the correct and incorrect answers, which sharpens the model's judgment and decision-making capabilities. Moreover, when combining the SFT and DPO, we observed the highest improvement in performance. This combination leverages the foundational alignment provided by SFT with the discriminative refinement offered by DPO, resulting in optimal trustworthiness and reliability in model outputs. The resulting model after SFT and DPO achieves 70.07 averaged score across all perspectives. These findings underscore the efficacy of our dual-phase training approach in MultiTrust, demonstrating it to be an effective strategy for developing reliable and safe LMs.

**The inference time router aligns the base LLM with the correct safety auxiliary model without additional training.** To validate the performance of the perplexity-based routing mechanism in MultiTrust, we conducted a comparative analysis between the performance of MultiTrust using our dynamic routing and that of MultiTrust using an oracle routing strategy. The oracle strategy represents an ideal scenario where the base model is aligned with the correct safety auxiliary model for each perspective during evaluation, serving as a potential upper bound for router performance. Results are detailed in Table 4. As illustrated in the table, the performance gap between MultiTrust and MultiTrust $_{oracle}$ is minimal, indicating the effectiveness of our routing approach. For example, the averaged performance of the MultiTrust-aligned Vicuna-7B model is 52.60, which closely

Table 3: **Adversarial robustness of the model after using different training methods**. For each method, we show the adversarial robustness scores of the model, including 3 subscores on 3 different testing sets: SST2, QQP, and MNLI, and an averaged score.

| Model | Adversarial Robustness | | | |
|---|---|---|---|---|
| | SST2 | QQP | MNLI | Avg |
| Llama-2-7B | 70.23 | 36.14 | 33.06 | 46.47 |
| Llama-2-7B (FT only) | 76.25 | 37.42 | 52.84 | 55.50 |
| Llama-2-7B (DPO only) | 75.31 | 50.07 | 68.96 | 64.78 |
| Llama-2-7B (FT + DPO) | **78.65** | **59.13** | **72.43** | **70.07** |

Table 4: **Trustworthiness performance of different routing mechanisms.** We show the trustworthiness performance of models on different perspectives. For adversarial robustness, we report 3 sub-scores on 3 different tasks: SST2, QQP, and MNLI. For fairness, we report 2 sub-scores on 2 different evaluation setting: Zero-shot and Few-shot. For truthfulness, we report 2 sub-scores: MC1 and MC2. All scores are the higher the better. We **bold** the highest score for each task and underline the second best. We find that the performance gap between MultiTrust and MultiTrust $_{oracle}$ is minimal, indicating the effectiveness of our routing mechanism. V-7: Vicuna-7B, L-13: Llama-2-13B, V-13: Vicuna-13B. ZS: Zero-shot. FS: Few-shot.

| Model | Adversarial Robustness | | | | Fairness | | | Truthfulness | | | Avg |
|---|---|---|---|---|---|---|---|---|---|---|---|
| | SST2 | QQP | MNLI | Avg | ZS | FS | Avg | MC1 | MC2 | Avg | |
| Vicuna-7B | 69.78 | 41.60 | 37.88 | 49.75 | 23.77 | 51.47 | 36.36 | 32.93 | 50.37 | 41.65 | 42.59 |
| MultiTrust $_{V-7}$ | 77.26 | 50.54 | 73.40 | 67.07 | 23.39 | 68.13 | 43.73 | 37.94 | 56.03 | 46.99 | 52.60 |
| MultiTrust $_{V-7, oracle}$ | 78.80 | **59.04** | 74.70 | 70.85 | 23.44 | **68.13** | 43.76 | 37.94 | 56.03 | 46.99 | 53.87 |
| Llama-2-13B | 52.55 | 38.30 | 21.04 | 37.30 | 21.81 | 47.13 | 33.32 | 28.03 | 43.95 | 35.99 | 35.54 |
| MultiTrust $_{L-13}$ | 75.22 | 48.22 | 71.13 | 64.86 | **26.27** | 65.40 | **44.06** | 35.13 | 53.89 | 44.51 | 51.14 |
| MultiTrust $_{L-13, oracle}$ | 78.20 | 55.89 | 73.33 | 69.13 | 26.25 | 65.20 | 43.96 | 35.13 | 53.89 | 44.51 | 52.53 |
| Vicuna-13B | 20.35 | 39.84 | 31.30 | 30.50 | -16.37 | 55.33 | 16.22 | 35.13 | 50.86 | 43.00 | 29.91 |
| MultiTrust $_{V-13}$ | 77.01 | 49.88 | 73.81 | 66.90 | 25.73 | 64.60 | 43.40 | 39.41 | 56.90 | 48.16 | 52.82 |
| MultiTrust $_{V-13, oracle}$ | **79.76** | 58.01 | **75.43** | **71.07** | 26.07 | 65.00 | 43.77 | **39.41** | **56.90** | **48.16** | **54.33** |

approaches the oracle performance of 53.25. Similarly, the Llama2-13B model under MultiTrust achieves a performance of 51.14, nearly matching the oracle result of 52.53. These findings confirm that the perplexity-based inference-time routing mechanism in MultiTrust is both effective and accurate, requiring no additional adaptation or training. This capability significantly enhances the flexibility in model alignment, ensuring that MultiTrust can dynamically adjust to varying trustworthiness requirements without compromising performance.

**Different perspectives have interactions and influence across each other.** In our analysis of Table 4, several interesting findings emerged regarding the interactions between different perspectives. Specifically, we observe cases where the performance under our dynamic routing surpassed that of oracle routing. This suggests that MultiTrust sometimes selects a different safety auxiliary model when evaluating on a specific perspective, and this alternate model can yield higher performance than the model specifically trained for that perspective. Such observations lead us to further investigate into the interactions among different trustworthiness perspectives. To systematically explore these relationships, we align a pre-trained Vicuna-7B model with various safety auxiliary models and evaluate each aligned configuration across all trustworthiness perspectives, as well as general benchmarks. The results are presented in Table 5. Our analysis reveals that, generally, the optimal performance for each perspective tends to be achieved when the model is aligned with the corresponding specialized safety auxiliary model. For instance, Vicuna-7B aligned with the robustness auxiliary model gets a robustness score of 70.85, and when aligned with the truthfulness auxiliary model, it achieves a truthfulness score of 46.99. Interestingly, we also observe cross-perspective benefits. Notably, the truthfulness auxiliary model improves performance in the fairness domain, achieving a score of 49.38. This indicates that certain model behaviors developed for one perspective can enhance performance in others. Overall, the Vicuna-7B model aligned with the adversarial robustness model achieves the highest cumulative score across trustworthiness and general tasks. This suggests that the data used to enhance robustness covers a broad spectrum of domains, thereby facilitating the development of a more universally trustworthy model without significant trade-offs in general performance.

Table 5: **Interactions and influence across different perspectives.** We report the performance of base model Vicuna-7B aligned with different safety auxiliary models. We report both trustworthiness performance and helpfulness. For trustworthiness, we report the scores on adversarial robustness, fairness, and truthfulness. For helpfulness, we report the scores on the ARC Challenge, HellaSwag, MMLU, and Winogrande. All scores are the higher the better. We **bold** the highest score of each task. We find that, generally, the optimal performance for each perspective is achieved when the model is aligned with the correct safety auxiliary model. We also cross-perspective benefits: the truthfulness auxiliary model improves the performance in fairness.

| Model | General performance | | | | Trustworthiness perspectives | | | |
|---|---|---|---|---|---|---|---|---|
| | ARC | HellaSwag | MMLU | Winogrande | Adv | Fair | Truth | Avg |
| Vicuna-7B | **53.92** | **77.43** | 49.97 | **72.38** | 49.75 | 36.36 | 41.65 | 42.59 |
| Vicuna-7B-Adv | 53.58 | 75.97 | **50.46** | 70.01 | **70.85** | 33.63 | 40.39 | **48.29** |
| Vicuna-7B-Fair | 53.67 | 75.85 | 50.12 | 68.82 | 44.58 | 43.76 | 39.56 | 42.63 |
| Vicuna-7B-Truth | 52.04 | 75.29 | 47.63 | 66.61 | 43.00 | **49.38** | **46.99** | 46.46 |

# 4 RELATED WORK

**Safety and trustworthiness in LLMs.** As LLMs are deployed across increasingly diverse domains, concerns are simultaneously growing about their trustworthiness. Zou et al. (2023) studies the safety of LLMs by introducing GCG algorithm to optimize a suffix string that can jailbreak LLMs. Chen et al. (2024) improves the model's trustworthiness by leveraging a gradual self-truthifying method to iteratively optimize the model. However, they only focus on a single specific trustworthiness perspective and can not extend to other perspectives. There are also many datasets and benchmarks evaluating the trustworthiness of LLMs. For example, AdvGLUE (Wang et al., 2021) and Prompt-Bench (Zhu et al., 2023) evaluate the adversarial robustness of language models. Recently, Liang et al. (2022) and Wang et al. (2023a) also focus on evaluating multiple perspectives of the model's trustworthiness.

**Multi-task learning** Existing work do multi-task learning either through model augmentation or model merging. Model augmentation improves the LM by incorporating or introducing new modules (Bansal et al., 2024), which is not flexible and requires re-training additional modules when augment with different models. Model merging aims to merge different models into a single model (Ilharco et al., 2022; Muqeeth et al., 2023; Ortiz-Jimenez et al., 2024). However, they require the LMs are trained from the base model and merged with it. They does not allow model merging with different initial parameters or different architectures. In MultiTrust, we do not require the safety auxiliary models to be trained from the base model, and we can augment models with different parameters and architectures, leading to better adaptability and real-world applications.

# 5 CONCLUSION

In this paper, we introduced MultiTrust, a novel framework designed to enhance the safety and trustworthiness of Large Language Models (LLMs) from multiple perspectives. By leveraging a perplexity-based routing mechanism, MultiTrust dynamically aligns the output of the base LLM with the outputs of the safety auxiliary models during inference, ensuring robust performance across a variety of trustworthiness metrics. Our experimental results demonstrate significant improvements in model trustworthiness without compromising on general performance, highlighting the framework's effectiveness and practical utility. Despite MultiTrust represents a significant advancement in the development of safer and more reliable LLMs, there are several limitations to consider. The current implementation relies heavily on the quality and diversity of the training data used for each safety model. In scenarios where data may be biased or insufficient, the effectiveness of the safety auxiliary models could be compromised. Additionally, the performance of the routing mechanism depends on the quality of the safety auxiliary models. The routing accuracy will be limited if the safety auxiliary models are not well-trained. The broader impacts of MultiTrust are twofold. By enhancing the safety and trustworthiness of LLMs, MultiTrust makes these models more viable for a range of critical applications, from automated decision-making in healthcare and finance to real-time monitoring systems in security-sensitive environments. Additionally, the framework encourages the adoption of new emerging trustworthiness perspectives into current LLMs, encouraging the community toward more responsible and sustainable AI development.

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

Table 6: **Detailed performance of MultiTrust on trustworthiness benchmarks.** We compare the performance of various open-access models with MultiTrust applied to three base LLMs. For each model, we report the detailed performance on trustworthiness, including sub-scores for each perspective. We report the scores on adversarial robustness, fairness, and truthfulness. For MultiTrust, we show the size of the base LLM. All scores are the higher the better. We **bold** the highest score of each task.

| Model | Size | Adversarial Robustness | | | | Fairness | | | Truthfulness | | |
|---|---|---|---|---|---|---|---|---|---|---|---|
| | | SST2 | QQP | MNLI | Avg | ZS | FS | Avg | MC1 | MC2 | Avg |
| Llama-2 | 7B | 70.23 | 36.14 | 33.06 | 46.47 | 6.24 | 48.27 | 25.34 | 30.23 | 45.32 | 37.78 |
| Gemma | 7B | 66.60 | 44.35 | 19.33 | 43.43 | 9.69 | 49.20 | 27.65 | 29.99 | 47.42 | 38.71 |
| Zephyr | 7B | 22.19 | 42.18 | 19.11 | 27.83 | 7.37 | 56.73 | 29.80 | 38.56 | 55.11 | 46.84 |
| Qwen v1.5 | 7B | 55.81 | 48.14 | 33.85 | 45.93 | 26.77 | 61.93 | 42.76 | 36.23 | 53.61 | 44.92 |
| Mistral v0.1 | 7B | 40.06 | 38.01 | 19.35 | 32.47 | **35.41** | 53.33 | 43.56 | 39.29 | 55.87 | 47.58 |
| Llama-3 | 8B | 63.89 | 44.09 | 19.86 | 42.61 | 20.25 | 58.20 | 37.50 | 36.35 | 51.63 | 43.99 |
| Vicuna | 33B | 69.19 | **50.62** | 65.96 | 61.92 | 32.86 | -0.57 | 17.66 | 37.21 | 56.03 | 46.62 |
| Vicuna | 7B | 69.78 | 41.60 | 37.88 | 49.75 | 23.77 | 51.47 | 36.36 | 32.93 | 50.37 | 41.65 |
| MultiTrust Vicuna-7B | 7B | **77.26** | 50.54 | 73.40 | 67.07 | 23.39 | **68.13** | 43.73 | 37.94 | 56.03 | 46.99 |
| Llama2 | 13B | 52.55 | 38.30 | 21.04 | 37.30 | 21.81 | 47.13 | 33.32 | 28.03 | 43.95 | 35.99 |
| MultiTrust Llama-2-13B | 13B | 75.22 | 48.22 | 71.13 | 64.86 | 26.27 | 65.40 | **44.06** | 35.13 | 53.89 | 44.51 |
| Vicuna | 13B | 20.35 | 39.84 | 31.30 | 30.50 | -16.37 | 55.33 | 16.22 | 35.13 | 50.86 | 43.00 |
| MultiTrust Vicuna-13B | 13B | 77.01 | 49.88 | **73.81** | **66.90** | 25.73 | 64.60 | 43.40 | **39.41** | **56.90** | **48.16** |

# A    DETAILED TRUSTWORTHINESS PERFORMANCE

We show the detailed performance of MultiTrust on trustworthiness benchmarks in Table 6. We compare the performance of various open-access models with MultiTrust applied to three base LLMs. For each model, we report the detailed performance on trustworthiness, including sub-scores for each perspective. We report the scores on adversarial robustness, fairness, and truthfulness. We can see consistent conclusions that MultiTrust brings significant improvements across all three trustworthiness perspectives when comparing the base models to their MultiTrust-aligned counterparts. When MultiTrust-aligned models are compared with other open-access models, it shows that MultiTrust not only enhances models of comparable sizes but also outperforms larger models.

