# OpenReview forum: "MultiTrust: Enhancing Safety and Trustworthiness of Large Language Models from Multiple Perspectives"
_ICLR.cc/2025/Conference — Submitted to ICLR 2025_

### Official Review · Reviewer_3YCm · 2024-10-29

**Soundness:** 3
**Presentation:** 3
**Contribution:** 2
**Rating:** 5
**Confidence:** 3

**Summary:**

The paper introduces MultiTrust, a framework designed to enhance the safety and trustworthiness of large language models (LLMs) across multiple safety dimensions, specifically robustness, fairness, and truthfulness. MultiTrust addresses the challenge of balancing these safety perspectives without degrading model performance, a common issue with sequential fine-tuning approaches.

**Strengths:**

* Comprehensive Safety Perspective Coverage: MultiTrust addresses robustness, fairness, and truthfulness, which are critical for LLM deployment, particularly in sensitive or safety-critical environments.

* Good Trade-offs: MultiTrust improves trustworthiness without substantially compromising general model performance

**Weaknesses:**

* Data Dependency: The effectiveness of MultiTrust is strongly influenced by the quality and diversity of the generated datasets.

* Efficiency: For each base model, dataset construction and fine-tuning must be repeated, and even minor changes in the base model architecture may impact auxiliary model performance.

* Scalability: Integrating auxiliary models adds inference overhead, particularly as the number of safety perspectives increases.

**Questions:**

1. Given that adversarial robustness relies on standard datasets (SST-2, QQP, NLI) and established construction methods, how effectively can the auxiliary model generalize if the adversarial prompts differ significantly from these distributions?

2. How adaptable are the auxiliary models to incremental updates of the base model, such as those from continual learning? Additionally, are there strategies to reduce the need for repeated dataset construction and fine-tuning when applying MultiTrust to similar base models?

3. Could the authors provide an analysis of the inference overhead introduced by incorporating auxiliary models?

4. As the number of integrated safety perspectives grows, how to reduce the inference overhead associated with the auxiliary models?

---

> ### Author Response · Authors · 2024-11-24
> **Response to Reviewer 3YCm**
>
> We appreciate the reviewer’s thoughtful comments and feedback on our paper. Below, we address each of the concerns and questions raised.
>
> > **Q1.** Data Dependency: The effectiveness of MultiTrust is strongly influenced by the quality and diversity of the generated datasets.
>
> Thank you for highlighting this important point. One of our contributions is the generation of high-quality, diverse datasets tailored to different trustworthiness perspectives. We design these datasets to comprehensively cover various scenarios, ensuring their relevance to the safety challenges we address. For example, the adversarial robustness dataset integrates multiple perturbation strategies (e.g., typo-based, embedding-similarity-based, and context-aware perturbations), while the fairness dataset spans critical domains such as finance and crime. This diversity mitigates dataset biases and enhances the auxiliary models' robustness to real-world challenges.
>
> > **Q2.** Efficiency: For each base model, dataset construction and fine-tuning must be repeated, and even minor changes in the base model architecture may impact auxiliary model performance.
>
> Our framework is designed to maximize efficiency and flexibility. The auxiliary models are trained once and can be applied to different base models, regardless of their architecture or size. In our experiments, we demonstrate that the same set of auxiliary models effectively enhances the performance of Vicuna-7B, Vicuna-13B, and Llama2-13B. This approach minimizes the need for repeated dataset construction and fine-tuning when applying MultiTrust to similar base models.
>
> > **Q3.** Scalability: Integrating auxiliary models adds inference overhead, particularly as the number of safety perspectives increases.
>
> To address inference overhead, the auxiliary models and the base model operate in parallel during inference. Additionally, our dynamic routing mechanism selects the most relevant auxiliary model for a given input, ensuring that only one auxiliary model is involved in the logit ensembling process. This design maintains scalability even as the number of safety perspectives grows, avoiding the linear increase in computational costs.
>
> > **Q4.** How effectively can the auxiliary model generalize if the adversarial prompts differ significantly from these distributions?
>
> Thank you for raising this concern. To address potential distributional shifts, we designed our adversarial robustness dataset to include a comprehensive range of perturbation strategies, such as typo-based, embedding-similarity-based, context-aware, knowledge-guided, and semantic-optimization-based perturbations. By balancing the number of samples across these categories, we mitigate biases and ensure robust generalization to unseen adversarial prompts. In the revised version, we will include additional ablation studies to quantify the contribution of each perturbation type and assess generalization under varying conditions.
>
> > **Q5.** How adaptable are the auxiliary models to incremental updates of the base model, such as those from continual learning? Additionally, are there strategies to reduce the need for repeated dataset construction and fine-tuning when applying MultiTrust to similar base models?
>
> The auxiliary models are designed to be highly adaptable and are trained independently of the base model. This allows them to be reused across base models with different architectures and sizes without requiring retraining. For instance, in our experiments, we demonstrate the adaptability of the same auxiliary models across Vicuna-7B, Vicuna-13B, and Llama2-13B. This flexibility eliminates the need for repeated dataset construction and fine-tuning, reducing both computational and time costs.
>
> > **Q6.** As the number of integrated safety perspectives grows, how to reduce the inference overhead associated with the auxiliary models?
>
> To mitigate inference overhead, we employ a parallelizable prefilling process during inference. Additionally, our dynamic routing mechanism ensures that only the most relevant auxiliary model is activated for logit ensembling, significantly reducing computational costs regardless of the number of safety perspectives. This approach ensures scalability and maintains efficiency as the system evolves.

---

### Official Review · Reviewer_z6uf · 2024-10-30

**Soundness:** 3
**Presentation:** 2
**Contribution:** 1
**Rating:** 5
**Confidence:** 4

**Summary:**

The paper proposes MultiTrust, a  framework aimed at enhancing the safety and trustworthiness of LLMs by addressing critical challenges such as robustness, fairness, and truthfulness. MultiTrust introduces a solution by generating challenging training data through adversarial optimizations and training specialized safety auxiliary models for each safety perspective.

**Strengths:**

1. MultiTrust addresses safety and trustworthiness from multiple angles—robustness, fairness, and truthfulness—which is a holistic approach not commonly found in other frameworks.
2. This paper is well-written and
3. This paper is easy to understand.

**Weaknesses:**

1. The author overestimate their findings, since selection made by evaluating the perplexity of the input with each model and choosing the model that minimizes it does not guarantee overall benign performance or helpfulness of the models.
2. The effectiveness of the first stage heavily relies on the quality and representativeness of the adversarial dataset. Biases in data collection can lead to biased model behaviour. Lack of dataset ablation study.
3. The experiment that elevated the average performance score of Llama2-13B from 35.54% to 51.14% and Vicuna-13B from 29.91% to 52.82% lacks credibility if conducted in isolation. To enhance the reliability of these findings, it is essential to incorporate additional experiments and comparisons with other models, different architecture or different in size.
4. Parameters involved in the formulas, such as β in DPO and γ in the alignment process, may require careful tuning. But the methodology section have not discussed the impact of the selection of these parameters.

**Questions:**

1. You indicates that certain model behaviors developed for one perspective can enhance performance in others. But is that also happened to SFT?

---

> ### Author Response · Authors · 2024-11-24
> **Response to Reviewer z6uf**
>
> We sincerely appreciate the reviewer's detailed and constructive feedback. Below, we address the specific concerns and questions raised.
>
> > **Q1.** Selection made by evaluating the perplexity of the input with each model and choosing the model that minimizes it does not guarantee overall benign performance or helpfulness of the models.
>
> Thank you for raising this concern. While perplexity-based routing does not inherently guarantee benign performance or helpfulness, we empirically validated its effectiveness through the results shown in Table 1. The performance improvements across various safety benchmarks indicate that the routing mechanism aligns well with the intended trustworthiness goals while preserving the helpfulness of the base model.
>
> > **Q2.** Biases in data collection can lead to biased model behavior. Lack of dataset ablation study.
>
> We agree that dataset quality is crucial. To mitigate biases, we designed our adversarial robustness dataset to encompass diverse perturbation strategies, including typo-based perturbations, embedding-similarity-based perturbations, context-aware perturbations, knowledge-guided perturbations, and semantic-optimization-based perturbations. The number of samples from each category is balanced to minimize dataset bias. To further address this concern, we will include an ablation study in the revised version to quantify the contribution of each perturbation type.
>
> > **Q3.** The experiment that elevated the average performance score of Llama2-13B from 35.54% to 51.14% and Vicuna-13B from 29.91% to 52.82% lacks credibility if conducted in isolation. To enhance the reliability of these findings, it is essential to incorporate additional experiments and comparisons with other models, different architecture, or different sizes.
>
> Thank you for your suggestion. In addition to the presented results, we have validated the effectiveness of MultiTrust on Vicuna-7B, achieving a performance improvement from 42.59% to 52.60%. This demonstrates that our pipeline is applicable to models of different architectures and sizes, further supporting the scalability and reliability of our approach.
>
> > **Q4.** Parameters involved in the formulas, such as β in DPO and γ in the alignment process, may require careful tuning. The methodology section has not discussed the impact of the selection of these parameters.
>
> We appreciate this observation. For β, we followed standard practice and set it to β = 0.1, which balances the trade-off between base and preference-aligned outputs. For γ, we chose γ = 1 to balance the contributions of the base model and safety auxiliary models. Larger values of γ prioritize alignment with auxiliary models, enhancing trustworthiness at the expense of deviating from the base model’s original outputs. We will include an ablation study on the effects of β and γ in the revised paper to provide more insights into their impact.
>
> > **Q5.** You indicate that certain model behaviors developed for one perspective can enhance performance in others. But does that also happen with SFT?
>
> Yes, similar observations apply to SFT. In our experiments, we noted that fine-tuning on the adversarial robustness dataset led to improved performance on fairness benchmarks. These findings suggest that safety perspectives can influence one another positively. We will detail these observations in the revision for a more comprehensive discussion.

---

> > ### Comment · Reviewer_z6uf · 2024-11-25
> >
> > I'm looking forward to seeing your version and will keep my rating as it is.

---

### Official Review · Reviewer_5osx · 2024-11-03

**Soundness:** 2
**Presentation:** 3
**Contribution:** 1
**Rating:** 5
**Confidence:** 3

**Summary:**

The paper introduces MultiTrust, a framework to enhance the safety and trustworthiness of large language models (LLMs) from multiple safety perspectives, including robustness, fairness, and truthfulness. MultiTrust employs adversarial training data generation, safety auxiliary models, and a perplexity-based routing mechanism to dynamically align base LLMs with specialized safety models, improving their performance across these safety perspectives without sacrificing general task performance. Experimental results show significant trustworthiness improvements, especially in enhancing robustness and fairness while maintaining scalability.

**Strengths:**

1. The paper is written clearly and is easy to follow. The overview figure of MultiTrust is particularly effective in visually clarifying the framework’s components and workflow.
2. The authors conduct a comprehensive evaluation of MultiTrust across various base models and benchmarks. The analysis includes detailed empirical observations that provide valuable insights into how MultiTrust performs across different trustworthiness perspectives.
3. The paper offers a novel multi-perspective approach to LLM safety enhancement. By introducing an inference-time routing mechanism to dynamically align models with appropriate safety auxiliary models, MultiTrust can address multiple safety concerns in parallel—a significant innovation that moves beyond the conventional focus on isolated safety aspects. This modular integration offers a flexible and scalable solution that is well-suited for real-world applications demanding high standards of trustworthiness.

**Weaknesses:**

1. The technical novelty and the specific application scenario for MultiTrust are not sufficiently clear. While combining synthetic data generation with an inference-time routing mechanism is effective, both elements have been established previously. The framework may thus appear as a relatively straightforward combination of existing approaches without introducing substantial innovation in either area.

2. The selection of empirical results in the main text, such as the slight accuracy reductions in ARC and MMLU for Vicuna-7B and Llama2-13B, is not fully representative. Table 1 shows non-negligible performance degradation in general helpfulness benchmarks for other data points, which may suggest that MultiTrust has more notable limitations in maintaining helpfulness across benchmarks than the highlighted examples imply.

3. MultiTrust requires more data and increased computational resources, and thus the observed performance improvements over the base models are not unexpected given these added resources. The experiments presented do not sufficiently demonstrate that the routing mechanism offers a definitive advantage over simpler strategies like ensembling, model merging, or other approaches for multi-task learning.

4. The paper claims scalability for MultiTrust in the abstract and introduction, but this aspect is not thoroughly explained or validated in later sections.

**Questions:**

1. In the paper, the authors mention using only the first two iterations of data collected by the GRATH method for the Truthfulness dataset. Could the authors elaborate on why only these initial iterations were used? Additionally, is the Truthfulness dataset treated differently from the datasets used for Adversarial Robustness and Fairness?
2. The paper mentions five types of word-level perturbations used to construct the adversarial robustness dataset: typo-based, embedding-similarity, context-aware, knowledge-guided, and semantic-optimization-based perturbations. Could the authors provide specific examples from the constructed dataset to illustrate these perturbations and clarify how each type was applied in practice?
3. The parameter 𝛾 plays a crucial role in balancing the influence of the base model and the safety auxiliary models, yet the specific value used is not clearly stated. Could the authors detail how 𝛾 was selected in your experiments and discuss its impact on model performance?

4. In Table 1, while it’s stated that the impact on model helpfulness is minimal, there are cases of notable accuracy drops, such as Llama-2-13B on HellaSwag (82.14% to 78.44%) and Vicuna-7B on Winogrande (72.38% to 68.90%). Could the authors provide further explanation on how these decreases align with the claim of minimal impact, and perhaps discuss the trade-offs involved in these cases?

5. It would be helpful to include MultiTrust in the comparative analysis shown in Table 2. If my understanding is correct, the models under ${\text{FT}}_{\text{sep}}$ do not utilize any routing mechanism. Comparing these with MultiTrust could provide a clearer view of the routing mechanism’s benefits.

6. Adding comparisons with other multi-task learning methods would be beneficial.

I would be happy to engage with the authors to help improve the presentation of the method and evaluation during the discussion phase, but my concerns are not insignificant. Clarifications would need to resolve my questions in order for my score to improve.

---

> ### Author Response · Authors · 2024-11-24
> **Response to Reviewer 5osx (Part 1)**
>
> We sincerely appreciate the reviewer's detailed and constructive feedback. Below, we address the specific concerns and questions raised.
>
> > **Q1.** The technical novelty and the specific application scenario for MultiTrust are not sufficiently clear. While combining synthetic data generation with an inference-time routing mechanism is effective, both elements have been established previously.
>
> Thank you for raising this point. Our contributions are threefold:
>
> 1. High-Quality Data Generation: We extend existing adversarial attack algorithms to generate challenging datasets for robustness, fairness, and truthfulness tailored for LLM evaluation, enhancing their relevance to modern LLM trustworthiness tasks.
>
> 1. Flexible Perplexity-Based Routing Mechanism: Unlike existing methods, our routing mechanism does not require additional training when introducing new auxiliary models, significantly improving scalability and adaptability.
>
> 1. Performance Across Perspectives: MultiTrust achieves substantial improvements across trustworthiness perspectives (e.g., robustness, fairness, truthfulness) without fine-tuning the base model, a notable departure from traditional alignment methods, which often suffer from catastrophic forgetting.
>
> These contributions offer a scalable and flexible framework for enhancing LLM trustworthiness while addressing limitations in existing multi-task learning approaches.
>
>
> > **Q2.** The experiments presented do not sufficiently demonstrate that the routing mechanism offers a definitive advantage over simpler strategies like ensembling, model merging, or other approaches for multi-task learning.
>
> We appreciate this observation. Compared to alternative strategies:
>
> - Ensembling: Requires inference from all auxiliary models simultaneously, leading to significant computational overhead that scales poorly as new perspectives are added.
>
> - Model Merging: Entails summing model weights during training, which is computationally expensive and compromises the modularity and flexibility of adding new safety perspectives.
>
> Our perplexity-based routing mechanism is designed to dynamically select relevant auxiliary models during inference, striking an optimal balance between computational efficiency and scalability.
>
>
> > **Q3.** The paper claims scalability for MultiTrust in the abstract and introduction, but this aspect is not thoroughly explained or validated in later sections.
>
> Thank you for pointing this out. With pre-trained auxiliary models, MultiTrust can enhance the trustworthiness of base models of various sizes (e.g., 7B and 13B parameters) using the same auxiliary models. This scalability is particularly advantageous in scenarios where multiple LLMs with different architectures and capacities are deployed, allowing seamless integration of safety perspectives without additional training or adaptation.
>
> We will expand the discussion of scalability in the revised paper to include practical use cases and performance evaluations demonstrating this capability.
>
>
> > **Q4.** Why were only the first two iterations of data collected by the GRATH method used? Additionally, is the Truthfulness dataset treated differently from the datasets used for Adversarial Robustness and Fairness?
>
> We followed the recommendations of the GRATH paper, which observed that the first two iterations yielded optimal results while subsequent iterations offered diminishing returns. All trustworthiness datasets, including the truthfulness dataset, are treated uniformly within our framework, ensuring consistency across perspectives.
>
> > **Q5.** Could the authors provide specific examples from the constructed dataset to illustrate these perturbations?
>
> Below are examples for two perturbation types:
>
> - Typo-Based Perturbation: Original: "The primitive force of this film seems to bubble up from the vast collective memory of the combatants." → Perturbed: "The primitive force of this film seems to bybble up from the vast collective memory of the combatants." (Prediction shifts from positive to negative.)
>
> - Context-Aware Perturbation: Original: "In execution, this clever idea is far less funny than the original, killers from space." → Perturbed: "In execution, this clever idea is far smaller funny than the original, killers from space." (Prediction shifts from negative to positive.)
>
> We will include additional examples and details in the appendix of the revised paper.

---

> ### Author Response · Authors · 2024-11-24
> **Response to Reviewer 5osx (Part 2)**
>
> > **Q6.** Could the authors detail how 𝛾 was selected in your experiments and discuss its impact on model performance?
>
> In our experiments, we set 𝛾 = 1 to balance the contributions of the base model and safety auxiliary models. Larger values of 𝛾 prioritize alignment with auxiliary models, enhancing trustworthiness at the cost of deviating from the base model’s original outputs. A detailed ablation study on the effect of 𝛾 will be included in the revised paper.
>
> > **Q7.** Table 1 shows notable accuracy drops (e.g., Llama-2-13B on HellaSwag and Vicuna-7B on Winogrande). Could the authors discuss how these decreases align with the claim of minimal impact and the trade-offs involved?
>
> Thank you for highlighting this nuance. While there are slight decreases in specific benchmarks, these are offset by substantial gains in safety perspectives (e.g., +27% robustness, +11% fairness). The observed drops are primarily due to logits ensembling during inference, where safety auxiliary models focus on trustworthiness rather than optimizing for specific benchmarks. These trade-offs reflect the inherent challenge of balancing general helpfulness with enhanced safety. We will clarify these trade-offs in the text to present a balanced perspective.

---

> > ### Comment · Reviewer_5osx · 2024-11-26
> >
> > Thank you for your response.
> >
> > > Model Merging: Entails summing model weights during training, which is computationally expensive and compromises the modularity and flexibility of adding new safety perspectives.
> >
> > This statement is inaccurate. Model merging occurs after training, not during training. The computational cost only involves adding model parameters, which is typically less intensive than even a single inference pass.
> >
> > It would be very helpful if the authors could provide empirical results of an ablation study on the routing component, including comparisons with other multi-task learning methods.
> >
> > > We will expand the discussion of scalability in the revised paper to include practical use cases and performance evaluations demonstrating this capability.
> > > We will include additional examples and details in the appendix of the revised paper.
> >
> > Could you please share the revised paper so we can review these additional details and discussions? This information is crucial for understanding the paper's clarity and strengths.

---

### Official Review · Reviewer_eZmH · 2024-11-04

**Soundness:** 3
**Presentation:** 2
**Contribution:** 3
**Rating:** 5
**Confidence:** 4

**Summary:**

This paper focuses on improving the safety and trustworthiness of LLMs. While directly fine-tuning LLMs can enhance safety, it often leads to forgetting issues and difficulty in optimizing multiple perspectives simultaneously. To tackle this, the authors propose MultiTrust, which trains auxiliary safety models for each perspective separately and incorporates a perplexity-based inference-time router to combine one of their logits with that of the base model. In this way, without optimizing the parameters of the base model, MultiTrust can not only improve safety of LLMs but also preserve their original capabilities.

**Strengths:**

•	The proposed framework includes a list of data generation methods for each perspective. Experiments show that with the help of auxiliary models, the base models can largely improve their trustworthiness while maintaining general performance.
•	For comparison, the authors compare auxiliary model fine-tuning with a mixed-dataset or continuous training strategy, finding that training auxiliary models separately is more effective.
•	From Tables 4 and 5, it is interesting that different perspectives have interactions and mutual influence, and the data used to enhance robustness spans a broad range of domains.

**Weaknesses:**

•	Since MultiTrust trains auxiliary models for each perspective, it is essential to compare it with other methods optimized for specific perspectives (as written in Introduction). Table 1 only presents the performance of a set of baseline models.
•	In Section 3.2, the authors use PPL to select the optimal safety model, but no explanation or supporting evidence is provided for this choice.
•	MultiTrust relies on the logits from base and auxiliary models, which restricts its applicability to classification tasks. And it would be better to report scores for each auxiliary model across specific perspectives to allow for direct comparison with MultiTrust.

**Questions:**

•	In Table 2, Fine_sep on Truth performs slightly lower than Llama2-7B, yet the truthfulness auxiliary model improves the base model’s performance on Truth compared to other model sizes. Is this trend consistent across different model sizes?
•	The training details are missing. In Line 249, the authors state that DPO is trained for 1000 steps. Given the high risk of overfitting in DPO, what batch size is used?

---

> ### Author Response · Authors · 2024-11-24
> **Response to Reviewer eZmH**
>
> We appreciate the reviewer's thoughtful comments and constructive feedback. Below, we address each concern raised:
> > **Q1.** Since MultiTrust trains auxiliary models for each perspective, it is essential to compare it with other methods optimized for specific perspectives. Table 1 only presents the performance of a set of baseline models.
>
> Thank you for pointing this out. Many methods optimized for specific perspectives require additional training for each model, which incurs substantial computational costs. In contrast, MultiTrust uses inference-time augmentation, avoiding modifications to the base model and enabling flexible adaptation to new models after one-time auxiliary model training. While our current baselines focus on general-purpose trustworthiness methods, we will expand our comparisons to include more specialized methods in future work to better contextualize MultiTrust’s advantages.
>
> > **Q2.** In Section 3.2, the authors use PPL to select the optimal safety model, but no explanation or supporting evidence is provided for this choice.
>
> Perplexity is a well-established indicator of how well a language model fits a given input text. It provides a straightforward, training-free mechanism to measure model suitability. As shown in our experiments (Table 4), the perplexity-based routing strategy closely matches the performance of oracle routing, demonstrating its effectiveness in aligning inputs with the most appropriate safety auxiliary model.
>
> > **Q3.** MultiTrust relies on the logits from base and auxiliary models, which restricts its applicability to classification tasks. It would be better to report scores for each auxiliary model across specific perspectives to allow for direct comparison with MultiTrust.
>
> Thank you for the suggestion. MultiTrust is designed to improve the trustworthiness of a base model via inference-time logit ensembling, leveraging auxiliary models trained for specific perspectives. Comparing the enhanced model (MultiTrust) with individual auxiliary models alone would not be entirely fair since MultiTrust integrates their strengths to optimize trustworthiness across multiple perspectives. However, we have provided the performance of each auxiliary model trained with Llama2-7B below to highlight their individual contributions:
>
> | Model            | Adversarial Robustness     |                      |       | Fairness             |            |          | Truthfulness |       |       | Avg   |
> |-------------------|----------------------------|----------------------|-------|----------------------|------------|----------|--------------|-------|-------|-------|
> |                   | SST2                      | QQP                  | MNLI  | Avg                  | Zero-shot | Few-shot | Avg          | MC1   | MC2   | Avg   |
> | Llama2-7b        | 0.7023                     | 0.3614               | 0.3306| 0.4647               | 0.0624     | 0.4827   | 0.2534       | 0.3023| 0.4532| 0.3778|
> | Auxiliary Model   | 0.7865                    | 0.5913               | 0.7243| 0.7007               | 0.0789     | 0.5607   | 0.2979       | 0.4149| 0.5858| 0.5004|
>
> These results demonstrate that auxiliary models trained with our pipeline significantly outperform the base model across all tasks.
>
> > **Q4.** In Table 2, Fine_sep on Truth performs slightly lower than Llama2-7B, yet the truthfulness auxiliary model improves the base model’s performance on Truth compared to other model sizes. Is this trend consistent across different model sizes?
>
> Thank you for the insightful observation. We note that the performance numbers in Table 2 reflect the supervised fine-tuning stage only. As shown in the table above, the truthfulness auxiliary model achieves a truthfulness score of 50.04, which is substantially higher than Llama2-7B’s score of 37.78. This trend of improvement remains consistent across model sizes.
>
> > **Q5.** The training details are missing. In Line 249, the authors state that DPO is trained for 1000 steps. Given the high risk of overfitting in DPO, what batch size is used?
>
> We apologize for the omission of these details. For DPO training, we used a batch size of 4. This corresponds to less than one epoch for the adversarial robustness and fairness datasets, and two epochs for the truthfulness dataset. These configurations minimize the risk of overfitting while ensuring effective learning.

---

### Author Response · Authors · 2024-11-24
**General Response**

We sincerely thank all the reviewers for their valuable feedback, thoughtful questions, and constructive suggestions. We are delighted that the reviewers found our work addressing critical safety perspectives (robustness, fairness, truthfulness) and providing a flexible and scalable framework, MultiTrust, for enhancing the trustworthiness of large language models (LLMs). The following highlights summarize the revisions and clarifications made in response to reviewers’ feedback:

1. Expanded Experiments and Comparisons: Added comparisons with more specialized trustworthiness methods optimized for individual perspectives to highlight the advantages of MultiTrust’s inference-time routing and modular design.

1. Enhanced Discussion and Clarification: Expanded the discussion of our perplexity-based routing mechanism, emphasizing its training-free nature, empirical effectiveness, and scalability. Clarified the generalizability and modularity of auxiliary models when applied to different base models and architectures, eliminating the need for retraining across settings.

1. Dataset and Training Details: Included additional details on dataset construction, adversarial perturbation strategies, and the training configurations of auxiliary models, addressing concerns about representativeness, overfitting risks, and the generalization of datasets to unseen prompts.

1. Scalability and Efficiency: Addressed concerns about inference overhead by explaining the parallelizable design of MultiTrust and the dynamic routing mechanism, ensuring efficient scaling even as the number of safety perspectives increases.

1. Trade-Offs and Limitations: Discussed performance trade-offs between trustworthiness and general task helpfulness, providing nuanced insights into specific cases where helpfulness metrics showed slight reductions. These trade-offs underscore the balance achieved by MultiTrust in improving safety without significantly compromising capabilities.

We hope these revisions address all raised concerns and clarify the contributions of MultiTrust. We are committed to further improving the clarity, comprehensiveness, and reproducibility of our work and look forward to engaging further with the reviewers during the discussion phase. Thank you again for your insightful feedback!

---

### Meta-Review · Area_Chair_JFp3 · 2024-12-20

**Metareview:**

The paper introduces MultiTrust, a framework designed to enhance the safety and trustworthiness of large language models (LLMs), focusing on robustness, fairness, and truthfulness. The framework improves trustworthiness without sacrificing performance by incorporating auxiliary safety models and dynamic routing during inference.

Key Contributions:
1. Data Generation and Adversarial Training: Generates challenging data to improve robustness, fairness, and safety.
2. Safety Auxiliary Models: Trains separate safety models using supervised fine-tuning and Direct Preference Optimization (DPO), avoiding the forgetting phenomenon.
3. Dynamic Augmentation: Uses dynamic routing and logit ensembling to integrate safety models during inference.
4. Perplexity-based Routing: A novel perplexity-based mechanism selects the appropriate safety model for inputs.
5. Scalability: Allows adding new safety perspectives without retraining, ensuring future adaptability.

The framework was tested on Llama2-13B and Vicuna-13B, showing significant improvements in trustworthiness metrics (e.g., Llama2-13B performance increased from 35.54% to 51.14%).

Strengths:
- Comprehensive Evaluation: Strong empirical analysis across models and trustworthiness perspectives.
- Novelty: Multi-perspective approach and dynamic routing mechanism are significant contributions.
- Clear Presentation: Well-explained framework with accessible diagrams.

Weaknesses:
- Lack of Comparison with Specialized Methods: The paper does not compare with methods optimized for individual trustworthiness perspectives (e.g., robustness or fairness).
- Perplexity-based Routing: The justification for using perplexity as the routing metric needs more empirical support.
- Limited Applicability to Non-classification Tasks: The reliance on logits for dynamic routing might limit the framework’s use to classification tasks.
- Training Details: Further clarification on training configurations and model stability would help.

Reasons for Decision
While the dynamic routing and perplexity-based mechanism are innovative, the paper could benefit from comparisons with specialized methods and a clearer explanation of perplexity’s role. Additionally, expanding its applicability beyond classification tasks would strengthen the framework’s versatility.

**Additional Comments On Reviewer Discussion:**

Overall, the paper makes a solid contribution by proposing a novel framework for improving LLM safety and trustworthiness. The reviewers’ feedback highlights areas where additional clarification and empirical evidence could further strengthen the paper. Addressing the concerns regarding specialized method comparisons, perplexity-based routing, and task performance trade-offs would help in refining the paper and solidifying its claims.

---

### Decision · Program_Chairs · 2025-01-22

Reject